# Bayesian Nonparametric Learning using the Maximum Mean Discrepancy Measure for Synthetic Data Generation

**Forough Fazeli Asl**[*]
Department of Mathematical and Statistical Sciences
University of Alberta
Edmonton, Canada
fazelias@ualberta.ca

**Michael Minyi Zhang**
Department of Statistics and Actuarial Science
University of Hong Kong
Hong kong, China
mzhang18@hku.hk

**Lizhen Lin**
Department of Mathematics
The University of Maryland
College Park, MD, USA
lizhen01@umd.edu

## Abstract

We introduce a Bayesian estimator for maximum mean discrepancy (MMD), enabling a novel approach to measure-based data generation. To demonstrate the adaptability of our method, we embed this estimator within a generative adversarial network (GAN) framework. This integration offers a powerful avenue for Bayesian nonparametric (BNP) learning, showcasing the estimator's broad applicability. Our BNP-driven GAN not only enhances sample diversity but also improves inferential accuracy, surpassing the performance of traditional methods. Further theoretical properties, proofs, and experiments are given by the Appendix.

## 1 Introduction

Data augmentation is the technique of generating synthetic data, often to train machine learning models when the data are scarce or the model is non-robust to perturbations in the data. When the likelihood is intractable to compute, evaluating the model's fit can be difficult. Maximum mean discrepancy (MMD) addresses this problem by enabling comparisons between distributions without explicit likelihoods through its feature matching properties. This property ensures that generated data matches the features of real data, making MMD an effective tool for evaluating and developing deep generative models. Bayesian nonparametric methods are a powerful tool with strong theoretical justifications but they have seen limited applications in MMD estimation. A key advantage of the Bayesian approach is that it incorporates expert knowledge through a prior distribution, offering regularization by introducing uncertainty in the sampling distribution via the Dirichlet process (DP). The absence of such methods in MMD estimation restricts statisticians who prefer Bayesian frameworks without making strong assumptions. This paper addresses this gap.

We propose a BNP estimator to accurately estimate the MMD between a parametric model and an unknown distribution by placing a DP prior on the unknown distribution. We extend the bootstrap method from [1] beyond posterior parameter inference, applying our estimator to training a generative adversarial network (GAN). Our approach uses the MMD estimator as a robust discriminator,

---

[*]Corresponding author

Workshop on Bayesian Decision-making and Uncertainty, 38th Conference on Neural Information Processing Systems (NeurIPS 2024).

combining MMD measurement with BNP inference to enhance GAN training, reduce mode collapse, and improve generator performance compared to frequentist (FNP) methods.

## 2 Previous work

Previous work on simulation-based inference has mainly used discrepancy measures from a frequentist nonparametric (FNP) perspective. Notably, GANs have been extensively explored in data augmentation and medical data synthesis where the fake images created by the GAN are used to supplement the training data [2–6]. A standard GAN features two neural networks: the generator $\{G_{\boldsymbol{\omega}}\}_{\boldsymbol{\omega} \in \mathcal{W}}$ and the discriminator $\{D_{\boldsymbol{\theta}}\}_{\boldsymbol{\theta} \in \Theta}$ [7]. The generator tries to fool the discriminator into misidentifying generated samples as real. However, these models are expanded beyond the classic loss function, which could potentially introduce challenges such as mode collapse–memorizing certain modes of data distribution while overlooking other diversities–and training instability.

A Bayesian approach, known as approximate Bayesian computation (ABC), estimates model parameters through simulation by comparing summary statistics of simulated and observed data [8]. ABC faces challenges in selecting informative summary statistics, which affects the accuracy of posterior inference [9, 10]. One particularly attractive choice of statistic is to use the MMD [11]. As the threshold decreases, ABC converges to the standard Bayesian posterior, which can be sensitive to model misspecification and lacks robustness [1].

To improve robustness, generalized Bayesian inference (GBI) replaces the likelihood with a robust loss function [12]. Example applications of GBI include using MMD in pseudo-likelihood approaches [13] or stochastic gradient MCMC for posterior inference [14]. Despite these advancements, GBI's sensitivity to hyperparameters and the computational demands of MCMC remain challenges [1]. To address these issues, an MMD posterior bootstrap method has been developed, offering a more efficient alternative [1, 15–17].

## 3 Dirichlet process

To perform Bayesian nonparametric learning (BNPL), we first must take samples from the Dirichlet process. The DP is an infinite generalization of the Dirichlet distribution that is considered on the sample space denoted as $\mathfrak{X}$, which possesses a $\sigma$-algebra $\mathcal{A}$ comprising subsets of $\mathfrak{X}$ [18]. $F$ follows a DP with parameters $(a, H)$ with the notation $F^{\text{Pri}} := (F \sim DP(a, H))$, if for any measurable partition $A_1, \ldots, A_k$ of $\mathfrak{X}$ with $k \geq 2$, the joint distribution of the vector $(F(A_1), \ldots, F(A_k))$ follows a Dirichlet distribution characterized by parameters $(aH(A_1), \ldots, aH(A_k))$. Moreover, it is assumed that $H(A_j) = 0$ implies $F(A_j) = 0$ with probability one. The base measure $H$ captures the prior knowledge regarding the data distribution, while $a$ signifies the strength or intensity of this knowledge.

As a conjugate prior, the posterior distribution of $F$ also follows a DP, denoted by $F^{\text{Pos}} := (F|\mathbf{X}_{1:n} \sim DP(a + n, H^*))$, for $n$ independent and identically distributed (IID) draws, $(\mathbf{X}_{1:n} \in \mathbb{R}^d)$, from the random probability measure $F$ where $H^* = a(a+n)^{-1}H + n(a+n)^{-1}F_{\mathbf{X}_{1:n}}$, and $F_{\mathbf{X}_{1:n}}$ represents the empirical cumulative distribution function of the sample $\mathbf{X}_{1:n}$.

To sample from the DP posterior, we use a finite approximation devised by Ishwaran and Zarepour [19], which allows for convenient simulation. In the context of posterior inference, this approximation is given by

$$F_N^{\text{Pos}} := \sum_{i=1}^{N} J_{i,N}^{\text{Pos}} \delta_{\mathbf{V}_i^{\text{Pos}}}, \tag{1}$$

where $\left(J_{1:N,N}^{\text{Pos}}\right) \sim \text{Dirichlet}((a + n)/N, \ldots, (a + n)/N)$, $\left(\mathbf{V}_{1:N}^{\text{Pos}}\right) \overset{\text{IID}}{\sim} H^*$, and $\delta_{\mathbf{V}^{\text{Pos}}}$ is the Dirac delta measure. In this study, the variables $J_{i,N}^{\text{Pos}}$ and $\mathbf{V}_i^{\text{Pos}}$ represent the DP's weight and location, respectively. The sequence $(F_N^{\text{Pos}})_{N \geq 1}$ converges in distribution to $F^{\text{Pos}}$, where $F_N^{\text{Pos}}$ and $F^{\text{Pos}}$ are random values in $M_1(\mathbb{R}^d)$, the space of probability measures on $\mathbb{R}^d$ endowed with the topology of weak convergence [19]. Although the stick-breaking representation is a commonly employed series representation for DP inference [20], it lacks the necessary normalization terms to convert it into a probability measure [21]. Additionally, simulating from an infinite series is only feasible through

using a random truncation approach to handle the terms within the series. In the subsequent sections, we investigate the efficacy of this approximation within a regularization method in a BNP generative model.

## 4 DPMMD-GAN: A Bayesian Nonparametric Learning in Data Generation

In our BNPL method, we define a DP prior on $F$, leading to a DP posterior on $F$ given the data. The key idea is that any posterior on the generator's parameter space $\mathcal{W}$ can be derived by mapping $F^{pos}$ approximately through the push-forward measure

$$\boldsymbol{\omega}^*(F^{pos}) := \arg\min_{\boldsymbol{\omega}\in\mathcal{W}} \mathrm{MMD}_{\mathrm{BNP}}(F_N^{pos}, F_{G_{\boldsymbol{\omega}}}), \tag{2}$$

where for a given sample $\mathbf{Y}_{1:m} \sim F_{G_{\boldsymbol{\omega}}}$ the posterior-based MMD estimator is defined by

$$\mathrm{MMD}_{\mathrm{BNP}}^2(F_{1,N}^{pos}, F_{2,m}) = \sum_{\ell,t=1}^{N} J_{\ell,N}^* J_{t,N}^* k(\mathbf{V}_\ell^*, \mathbf{V}_t^*) - \frac{2}{m}\sum_{\ell=1}^{N}\sum_{t=1}^{m} J_{\ell,N}^* k(\mathbf{V}_\ell^*, \mathbf{Y}_t) + \frac{1}{m^2}\sum_{\ell,t=1}^{m} k(\mathbf{Y}_\ell, \mathbf{Y}_t). \tag{3}$$

In this context, the discriminator $D$ can be viewed as a black box that uses the MMD estimator to differentiate between the real and fake data, reducing the computational cost compared to a neural network-based discriminator. We discuss some of the properties of estimator (3) in the Appendix. Let $\boldsymbol{\omega}^*$ be the optimal parameter of $G_{\boldsymbol{\omega}}$ that minimizes $\mathrm{MMD}_{\mathrm{BNP}}(F_N^{pos}, F_{G_{\boldsymbol{\omega}},m})$. Since $\mathrm{MMD}_{\mathrm{BNP}}(F_N^{pos}, F_{G_{\boldsymbol{\omega}},m})$ serves as a BNP estimation of (5), it is crucial to evaluate this estimation's accuracy, focusing on the GAN's ability to generate realistic samples (generalization error) and handle outliers (robustness). Lemma 2 in the appendix addresses these aspects. While the previous statements provide upper bounds for the MMD estimator's expectation, the next lemma offers stochastic bounds on the estimation error to assess posterior consistency.

**Lemma 1** *Building upon the general assumptions stated in Lemma 2, for a given sample* $\mathbf{X}_1, \ldots, \mathbf{X}_n$ *from distribution $F$ in the probability space $(\mathfrak{X}, \mathcal{A}, \mathrm{Pr})$ and any $\epsilon > 0$,*
*i.* $\quad \mathrm{Pr}\left(|\mathrm{MMD}(F_N^{pos}, F_{G_{\boldsymbol{\omega}^*},m}) - \mathrm{MMD}(F, F_{G_{\boldsymbol{\omega}'}})| \geq h(n,m,K,\epsilon) + |\Delta_1| + |\Delta_2|\right) \quad \leq$
$2\exp\frac{-\epsilon^2 nm}{2K(n+m)},$

*ii.* $\mathrm{Pr}\left(\mathrm{MMD}(F, F_{G_{\boldsymbol{\omega}^*}}) > \epsilon\right) \leq \frac{1}{\epsilon}\left(\mathrm{MMD}(F, F_{G_{\boldsymbol{\omega}'}}) + \frac{2K}{\sqrt{n}} + \frac{4aK}{a+n} + 2\sqrt{\frac{(a+n+N)K}{(a+n+1)N}}.\right),$
*where,* $h(n,m,K,\epsilon) = 2\sqrt{K}(\sqrt{n} + \sqrt{m})/\sqrt{nm} + \epsilon$, $\Delta_1 = \mathrm{MMD}(F_N^{pos}, F_{G_{\boldsymbol{\omega}^*}}) - \mathrm{MMD}(F_n, F_{G_{\boldsymbol{\omega}'},m})$, *and* $\Delta_2 = \mathrm{MMD}(F, F_{G_{\boldsymbol{\omega}^*}}) - \mathrm{MMD}(F, F_{G_{\boldsymbol{\omega}'}})$.

A direct consequence of Lemma 1(ii) is that for a fixed value of $a$, $\mathrm{Pr}(\mathrm{MMD}(F, F_{G_{\boldsymbol{\omega}^*}}) \geq \epsilon) \to 0$, as $n \to \infty$ and $N \to \infty$, for any $\epsilon > 0$, when $\mathrm{MMD}(F, F_{G_{\boldsymbol{\omega}'}}) = 0$ (well-specified case). This implies $F_{G_{\boldsymbol{\omega}^*}}$ converges in probability to the data distribution $F$ as the sample size increases in well-specified cases. A detailed guide on choosing DP hyperparameters and kernel settings is provided in the Appendix.

## 5 Experimental results

We consider the MNIST dataset including handwritten digits with 10 modes, bone marrow biopsy (BMB) histopathology, Labeled Faces in the Wild (LFW) Dataset, and brain MRI images to analyze the model performance. All data description are given by the Appendix. Following the design choices of [22], we use the Gaussian neural network for the generator with four hidden layers each having rectified linear units activation function and a sigmoid function for the output layer. We also set mini-batch sizes to be $n_{mb} = 1,000$ and use a mixture of six Gaussian kernels corresponding to the bandwidth parameters $2, 5, 10, 20, 40$, and $80$ to train networks in $40,000$ iterations. We generate samples from the trained BNP GAN using Algorithm 2 from the Appendix, as depicted in Figure 1-Row 2. The results of [22] are also presented by Figure 1-Row 3, as the frequentist counterpart of our BNP procedure. Based on these preliminary results, we can see that our generated images can, at least, replicate the results of [22] and in some cases produce sharper images. This result can also be deduced from the presented scores of MMD, Kernel Inception Distance (KID) [23] and Frechet

Inception Distance (FID) [24] in Table 1. The low scores suggest better performance, as smaller values indicate closer similarity to real images.

Conversely, our results show that the BNP GAN with a mixture of Gaussian kernels outperforms the single Gaussian kernel approach. To explore this further, Figure 2 in the Appendix presents samples from the trained generator using various $\sigma$ values and the median heuristic $\sigma_{MH}$. Note that $\sigma_{MH}$ is updated in each iteration, so no specific value is reported. Although higher $\sigma$ increases image diversity, the resolution remains below that achieved with the mixture kernel.

To assess whether the proposed discriminator used in the BNP GAN leads to faster or better convergence of the generated samples compared to the baseline proposed by [22], we consider the synthetic distribution $\frac{1}{2}N(-\mathbf{1}_d, I_d) + \frac{1}{2}N(\mathbf{1}_d, I_d)$ as the true distribution and provide the corresponding MMD values for both models over 20,000 iterations in the data generation process, as shown in Figure 3 of Appendix. Our proposed GAN clearly displays a higher speed of convergence for the corresponding cost function to zero, and thus better performance compared to the baseline.

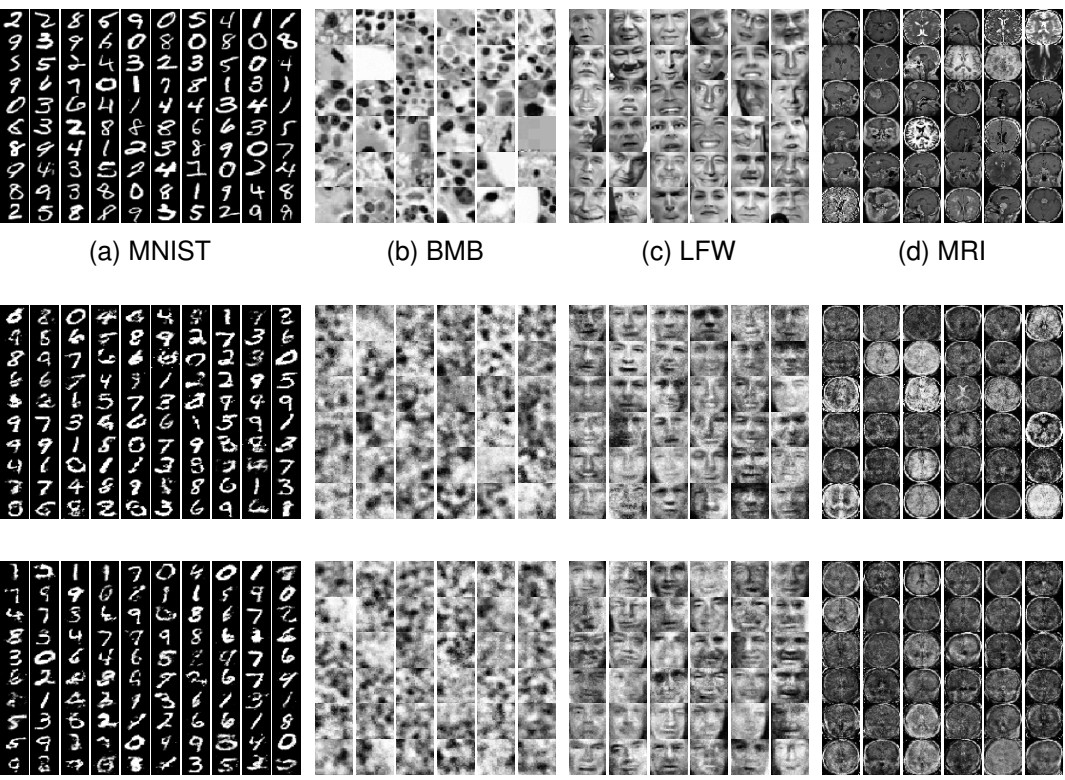

|  (a) MNIST | (b) BMB | (c) LFW | (d) MRI |

Figure 1: Generated samples from training dataset (Row 1), BNP model (Row 2), and FNP model (Row 3).

Table 1: The values of MMD, KID, and FID scores for four groups of datasets.

| Scores | Dataset | | | | | | | |
|--------|---------|---|---|---|---|---|---|---|
| | MNIST | | BMB | | LFW | | MRI | |
| | BNP | FNP | BNP | FNP | BNP | FNP | BNP | FNP |
| MMD | 0.0384 | 0.0404 | 0.0285 | 0.0315 | 0.0281 | 0.0302 | 0.2059 | 0.2231 |
| KID | 0.0034 | 0.0046 | 0.0030 | 0.0036 | 0.0019 | 0.0026 | 0.0260 | 0.0264 |
| FID | 35.560 | 37.934 | 17.006 | 17.264 | 14.010 | 14.473 | 87.975 | 87.831 |

# 6    Concluding remarks

Our BNP approach effectively estimates the MMD between an unknown and an intractable parametric distribution, showing promise in training GANs by using the estimator as a discriminator to induce a posterior on the generator's parameters. The stick-breaking representation, however, lacks normalization and shows stochastic decrease, making it inefficient for simulating from a DP [21]. Exploring alternative DP approximations for MMD estimation is a promising direction for future research. Future work will also focus on generating 3D medical images to further improve results.

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

# Appendix

## A  Maximum mean discrepancy distance

For a given data space $\mathfrak{X}$, consider the random variables $\mathbf{X}$ and $\mathbf{Y}$, drawn from distributions $F_1$ and $F_2$ respectively. Here, $F_1$ and $F_2$ belong to $\mathcal{B}(\mathfrak{X})$, which represents the set of Borel probability distributions on $\mathfrak{X}$. We consider the discrepancy $d : \mathcal{B}(\mathfrak{X}) \times \mathcal{B}(\mathfrak{X}) \to [0, \infty)$ through the integral pseudo-probability metric (IPM) [25], defined as shown in (4). The class of functions $\mathcal{F}$ is designed to be rich enough to distinguish between $F_1$ and $F_2$, and restrictive enough to provide accurate estimates based on a finite sample.

$$d_{\text{IPM}}(F_1, F_2) = \sup_{h \in \mathcal{F}} |E_{F_1}(h(\mathbf{X})) - E_{F_2}(h(\mathbf{Y})))|. \tag{4}$$

The MMD is then defined by considering $\mathcal{F} = \{h \in \mathcal{H}_k | \, ||h||_{\mathcal{H}_k} \leq 1\}$, which represents a unit ball in a reproducing kernel Hilbert space (RKHS) $\mathcal{H}_k$ with associated kernel $k : \mathfrak{X} \times \mathfrak{X} \to \mathbb{R}$. In this context, $||\cdot||_{\mathcal{H}_k}$ denotes the norm function in the RKHS. The function $k(\cdot, \cdot)$ is positive definite, such that for any function $h \in \mathcal{H}_k$ and any $\mathbf{X} \in \mathfrak{X}$, $h(\mathbf{X}) = \langle h, k(\mathbf{X}, \cdot) \rangle_{\mathcal{H}_k}$, where $\langle \cdot, \cdot \rangle_{\mathcal{H}_k}$ represents the inner product in $\mathcal{H}_k$. Consider function $\mu_{F_1}(\cdot) = E_{F_1}[k(\mathbf{X}, \cdot)] \in \mathcal{H}_k$, which is defined as the kernel mean embedding of the distribution $F_1$ in [26]. Then, for given $\mathbf{X}, \mathbf{X}' \overset{i.i.d.}{\sim} F_1, \mathbf{Y}, \mathbf{Y}' \overset{i.i.d.}{\sim} F_2$, if $E_F(\sqrt{k(\mathbf{X}, \mathbf{X})}) < \infty$ for all $F \in \mathcal{B}(\mathfrak{X})$, the MMD is given by

$$\text{MMD}^2(F_1, F_2) = ||\mu_{F_1} - \mu_{F_2}||^2_{\mathcal{H}_k} = E_{F_1}[k(\mathbf{X}, \mathbf{X}')] - 2E_{F_1, F_2}[k(\mathbf{X}, \mathbf{Y})] + E_{F_2}[k(\mathbf{Y}, \mathbf{Y}')]. \tag{5}$$

Note that $\text{MMD}^2(F_1, F_2) = 0$ if and only if $F_1 = F_2$, when $\mathcal{H}_k$ is a *universal* RKHS defined on a *compact* metric space $\mathfrak{X}$ and $k(\cdot, \cdot)$ is *continuous* [26, Theorem 5]. In practice, distributions $F_1$ and $F_2$ are not accessible, and then the biased, empirical estimator of (5) (V-statistic) is calculated using empirical distributions $F_{1,n}$ and $F_{2,m}$ as

$$\text{MMD}^2(F_{1,n}, F_{2,m}) = \frac{1}{n^2} \sum_{i,j=1}^{n} k(\mathbf{X}_i, \mathbf{X}_j) - \frac{2}{mn} \sum_{i=1}^{n} \sum_{j=1}^{m} k(\mathbf{X}_i, \mathbf{Y}_j) + \frac{1}{m^2} \sum_{i,j=1}^{m} k(\mathbf{Y}_i, \mathbf{Y}_j), \tag{6}$$

where $\mathbf{X}_1, \ldots, \mathbf{X}_n$ is a sample from $F_1$ and $\mathbf{Y}_1, \ldots, \mathbf{Y}_m$ is a sample generated from $F_2$.

## B  Choosing DP hyperparameters

In the context of approximating the posterior on the parameter space, the prior choice for $F$ and determining the strength of belief becomes challenging. We consider a small value for $a$ as a non-informative prior, following the suggestion by [1], thanks to its broad ability to characterize uncertainty [27]. However, it's important to note that setting $a = 0$ as done by [1] is not always well-defined mathematically, as the DP is only defined for $a > 0$. Therefore, we opt for $a = 10^{-6}$. In this case, the DP posterior remains invariant to the choice of $H$.

## C  Kernel Settings

In our method, we choose to use the standard radial basis function (RBF) kernel as its feature space corresponds to a universal RKHS. [22, 28] and [29] used the Gaussian kernel in training MMD-GANs because of its simplicity and good performance. [28] also evaluated some other RBF kernels such as the Laplacian and rational quadratic kernels to compare the results of the MMD-GANs with those obtained based on using Gaussian kernels. They found the best performance by applying the Gaussian kernel in the MMD cost function.

Hence, we consider the Gaussian kernel function in our proposed procedure. To choose the bandwidth parameter $\sigma$, we follow the idea of considering a set of fixed values of $\sigma$'s such as $\{\sigma_1, \ldots, \sigma_T\}$, then compute the mixture of Gaussian kernels $k(\cdot, \cdot) = \sum_{t=1}^{T} k_{G_{\sigma_t}}(\cdot, \cdot)$, to consider in (3). For each $\sigma(t)$, $0 \leq k_{G_{\sigma_t}}(\cdot, \cdot) \leq 1$; hence, $0 \leq k(\cdot, \cdot) \leq T$, which satisfies the theoretical results presented in the paper. As it is mentioned in [22], this choice reflects a good performance in training MMD-GANs.

## C.1 Radial Basis Function Kernels Family

The construction of MMD-based procedures is proposed based on considering a kernel function with feature space corresponding to a universal RKHS. The radial basis function (RBF) kernel is the most well-known kernel family satisfying the above situation. For two vectors $\mathbf{X}, \mathbf{Y} \in \mathbb{R}^d$, the RBF kernel is represented by

$$k(\mathbf{X}, \mathbf{Y}) = h(||\mathbf{X} - \mathbf{Y}||/\sigma),$$

where, $h$ is a function from the positive real numbers $\mathbb{R}^+$ to $\mathbb{R}^+$, $|| \cdot ||$ represents the $L^2$-norm, and $\sigma$ is the bandwidth parameter that indicates the kernel size. There are many functions assigned to $h$, for example, the Gaussian, exponential, rational quadratic kernels, and Matern, represented by

$$h_1(x) = \exp\left(-\frac{x^2}{2}\right), \; h_2(x) = \exp\left(-x\right), \; h_3(x) = \left(1 + \frac{x^2}{2\alpha}\right)^{-\alpha}, \; h_4(x) = (1 + \sqrt{2\nu}x)e^{-\sqrt{2\nu}x},$$

respectively; where, $\alpha$ in $h_3$ is a positive-valued scale-mixture parameter, and the $\nu$ in $h_4$ is a parameter that controls the smoothness of the kernel results [30, 31].

One of the simplest kernel functions above is the Gaussian kernel, which is mostly used in machine learning problems and only depends on bandwidth parameter $\sigma$. The Gaussian kernel tends to 0 and 1 when $\sigma \to 0$ and $\sigma \to \infty$, respectively. Both situations lead to $\mathrm{MMD}^2$ being zero. Hence, the choice of the parameter $\sigma$ has a crucial effect on the performance of this kernel. Numerous methods are proposed to choose the value of $\sigma$, however, there is no definitive optimization method for this problem. The median heuristic is one of the first methods used in choosing $\sigma$ empirically and will be denoted in our experimental results by $\sigma_{MH}$. More precisely, for two samples $\{\mathbf{X}_i\}_{i=1}^n$ and $\{\mathbf{Y}_i\}_{i=1}^m$, the $\sigma_{MH}$ is considered as the median of $\{||\mathbf{X}_i - \mathbf{Y}_j||^2 : 1 \leq i \leq n, 1 \leq j \leq m\}$, which is mostly used in kernel-based tests [32]. Selecting $\sigma$ based on maximizing the power of two-sample problems is another strategy considered by [33]. The selection of the MMD bandwidth on held-out data to maximize power was first proposed by [34] for linear-time estimates and by [35] for quadratic-time estimates. Recently, bandwidth selection without data splitting has been proposed for quadratic [36] and linear [37] MMD estimates. Regarding the choice of $\sigma$ in kernel-based GANs, a common idea is assigning several fixed values to $\sigma$ and then considering the mixture of their corresponding Gaussian kernel. This strategy has received much attention and shown an acceptable performance in training GANs[2].

# D  Computational Algorithm

## D.1  Training the BNP GAN

---

**Algorithm 1** Pseudocode of training a GAN using the BNP approach

---

1: Set $a = 10^{-6}$ to employ a non-informative prior leading DP posterior $DP(n, F_n)$.
2: Initialize $N$.
3: $r_{mn} \leftarrow$ Number of training iteration, $n_{mb} \leftarrow$ Mini-batch size
4: $\boldsymbol{\omega}_0 \leftarrow$ An initial parameter for generator $G_{\boldsymbol{\omega}}$, $\{\mathbf{x}_\ell\}_{\ell=1}^n \leftarrow$ real dataset
5: **for** $i \leftarrow 0$ to $r_{mb}$ **do**
6:      Generate a random sample $\{\mathbf{x}_\ell^{mb}\}_{\ell=1}^{n_{mb}}$ from real dataset $\{\mathbf{x}_\ell\}_{\ell=1}^n$
7:      Generate a sample of noise vector $\{\mathbf{u}_\ell\}_{\ell=1}^{n_{mb}}$ from uniform distribution $U(-1, 1)$
8:      Generate a sample from $F_{G_{\boldsymbol{\omega}_i}}$, distribution of $G_{\boldsymbol{\omega}_i}$, as $\{\mathbf{y}_\ell = G_{\boldsymbol{\omega}_i}(\mathbf{u}_\ell)\}_{\ell=1}^{n_{mb}}$
9:      Generate a sample of size $N$ from $F^{pos} = F|\{\mathbf{x}_\ell^{mb}\}_{\ell=1}^{n_{mb}}$ using $\sum_{i=1}^N J_{i,N}^* \delta_{\mathbf{v}_i^*}$.
10:      Use generated samples in steps 9 and 10 to compute $\mathrm{MMD}_{\mathrm{BNP}}^2(F_N^{pos}, F_{G_{\boldsymbol{\omega}_i}, N})$.
11:      Compute the gradient:

$$\frac{\partial \mathrm{MMD}_{\mathrm{BNP}}(F_N^{pos}, F_{G_{\boldsymbol{\omega}_i}, m})}{\partial \boldsymbol{\omega}_i} = \frac{1}{2\sqrt{\mathrm{MMD}_{\mathrm{BNP}}^2(F_N^{pos}, F_{G_{\boldsymbol{\omega}}, m})}} \frac{\partial \mathrm{MMD}_{\mathrm{BNP}}^2(F_N^{pos}, F_{G_{\boldsymbol{\omega}}, m})}{\partial \boldsymbol{\omega}}.$$

12:      Use backpropagation for calculating partial derivatives $\frac{\partial \mathbf{G}_{\boldsymbol{\omega}_i}(\mathbf{u}_\ell)}{\partial \boldsymbol{\omega}_i}$ in the previous step to update parameter $\boldsymbol{\omega}_i$.
13: **end for**
14: **return** $\boldsymbol{\omega}^*$          ▷ An optimized parameter for $G_{\boldsymbol{\omega}}$ that minimizes the cost function.

---

[2]For further details, see [22] and [29].

# E Theoretical proofs

**Proposition 1** *For a non-negative real value $a$ and fixed probability distribution $H$, let $F_1^{pri} := F_1 \sim DP(a, H)$ and $(J_{1,N}, \ldots, J_{N,N}) \sim Dirichlet(\frac{a}{N}, \ldots, \frac{a}{N})$ be the weights in the approximation of $F^{pri}$, given by [19]. Then, as $a \to \infty$,*
*i. $J_{\ell,N} \xrightarrow{a.s.} \frac{1}{N}$, for any $\ell \in \{1, \ldots, N\}$,*
*ii. $J_{\ell,N} J_{t,N} \xrightarrow{a.s.} \frac{1}{N^2}$, for any $\ell, t \in \{1, \ldots, N\}$, where $\ell \neq t$.*

**Proof of Proposition 1** *Recall*

$$F_N^{pri} = \sum_{i=1}^{N} J_{i,N} \delta_{Y_i}. \tag{7}$$

*Since $E_{F_1^{pri}}(J_{\ell,N}) = \frac{1}{N}$, for any $\ell \in \{1, \ldots, N\}$ and $\epsilon > 0$, Chebyshev's inequality implies*

$$\Pr\{|J_{\ell,N} - 1/N| \geq \epsilon\} \leq \frac{Var(J_{\ell,N})}{\epsilon^2},$$

*where, $Var_{F_1^{pri}}(J_{\ell,N}) = \frac{N-1}{N^2(a+1)}$. Assuming $a = \kappa^2 c$ for $\kappa \in \mathbb{N}$ and a fixed positive number $c$, gives*

$$\Pr\{|J_{\ell,N} - 1/N| \geq \epsilon\} \leq \frac{1}{\kappa^2 c \epsilon^2}.$$

*The convergence of series $\sum_{\kappa=0}^{\infty} \kappa^{-2}$ implies $\sum_{\kappa=0}^{\infty} \Pr\{|J_{\ell,N} - 1/N| \geq \epsilon\} < \infty$. By letting $a \to \infty$, the first Borel Cantelli lemma concludes $|J_{\ell,N} - 1/N| \xrightarrow{a.s.} 0$ and the result of (i) follows. To prove (ii), it is enough to show $\Pr\{\lim_{a \to \infty}(J_{\ell,N} J_{t,N}) \neq \frac{1}{N^2}\} = 0$. To prove this for the probability space $(\Omega, \mathcal{F}, \Pr)$, let*

$$A = \left\{\omega \in \Omega : \lim_{a \to \infty} (J_{\ell,N}(\omega) J_{t,N}(\omega)) \neq \frac{1}{N^2}\right\}, \quad B = \left\{\omega \in \Omega : \lim_{a \to \infty} (J_{\ell,N}(\omega)) \neq \frac{1}{N}\right\},$$

$$C = \left\{\omega \in \Omega : \lim_{a \to \infty} (J_{t,N}(\omega)) \neq \frac{1}{N}\right\},$$

*where, $\Pr(B)$ and $\Pr(C)$ are zero by (i). Since $A \subseteq B \cup C$, then,*

$$1 - \Pr\left\{\omega \in \Omega : \lim_{a \to \infty} (J_{\ell,N}(\omega) J_{t,N}(\omega)) = \frac{1}{N^2}\right\} = \Pr(A) \leq \Pr(B) + \Pr(C) = 0,$$

*which concludes the result.*

**Theorem 1** *For a non-negative real value $a$ and fixed probability distribution $H$, let $F_1^{pri} := (F_1 \sim DP(a, H))$ and $k(\cdot, \cdot)$ be any continuous kernel function with feature space corresponding to a universal RKHS defined on a compact metric space $\mathfrak{X}$. Assume that $|k(\boldsymbol{z}, \boldsymbol{z}')| < K$, for any $\boldsymbol{z}, \boldsymbol{z}' \in \mathbb{R}^d$. Then, for a given sample $\mathbf{X}_1, \ldots, \mathbf{X}_n$ from distribution $F_1$,*
*i. as $a \to \infty$ (informative prior),*

    *a. $\mathrm{MMD}^2_{\mathrm{BNP}}(F_{1,N}^{pos}, F_{2,m}) \xrightarrow{a.s.} \mathrm{MMD}^2(H_N, F_{2,m})$,*

    *b. $E(\mathrm{MMD}^2_{\mathrm{BNP}}(F_{1,N}^{pos}, F_{2,m})) \to \mathrm{MMD}^2(H, F_2)$, $N \to \infty$, and $m \to \infty$,*

*ii. as $n \to \infty$ (consistency),*

    *a. $\mathrm{MMD}^2_{\mathrm{BNP}}(F_{1,N}^{pos}, F_{2,m}) \xrightarrow{a.s.} \mathrm{MMD}^2(F_{1,N}, F_{2,m})$,*

    *b. $E(\mathrm{MMD}^2_{\mathrm{BNP}}(F_{1,N}^{pos}, F_{2,m})) \to \mathrm{MMD}^2(F_1, F_2)$, as $N \to \infty$, $n \to \infty$, and $m \to \infty$.*

**Proof of Theorem 1** *For samples $\{\mathbf{V}_\ell\}_{\ell=1}^N$ and $\{\mathbf{Y}_\ell\}_{\ell=1}^m$, respectively, from $H$ and $F_2$, the triangle inequality implies*

$$\left| \text{MMD}^2_{\text{BNP}}(F_{1,N}^{pri}, F_{2,m}) - \text{MMD}^2(H_N, F_{2,m}) \right| \le K \Bigg\{ \sum_{\ell,t=1}^N \left| J_{\ell,N} J_{t,N} - \frac{1}{N^2} \right|$$

$$+ \frac{2}{m} \sum_{\ell=1}^N \sum_{t=1}^m \left| J_{\ell,N} - \frac{1}{N} \right| \Bigg\}.$$

*By Proposition 1, which provides some theoretical properties of the DP approximation given in (7), the right-hand side of the above inequality converges almost surely to 0 as $a \to \infty$ for fixed $N$. This convergence immediately concludes the proof of (i). To prove (ii), since $(J_{1,N}, \ldots, J_{N,N}) \sim \text{Dirichlet}(\frac{a}{N}, \ldots, \frac{a}{N})$, $E_{F_1^{pri}}(J_{\ell,N}) = \frac{1}{N}$ and*

$$E_{F_1^{pri}}(J_{\ell,N} J_{t,N}) = \begin{cases} \dfrac{a}{(a+1)N^2} & \text{if } \ell \ne t, \\ \dfrac{a+N}{(a+1)N^2} & \text{if } \ell = t. \end{cases}$$

*Applying these properties in definition of $\text{MMD}^2_{\text{BNP}}(F_{1,N}^{pri}, F_{2,m})$ results in*

$$E_{F_1^{pri}}(\text{MMD}^2_{\text{BNP}}(F_{1,N}^{pri}, F_{2,m}) | \mathbf{V}_{1:N}) = \sum_{\ell=1}^N \sum_{t \ne \ell}^N \frac{a k(\mathbf{V}_\ell, \mathbf{V}_t)}{(a+1)N^2} + \sum_{\ell=1}^N \sum_{t=\ell}^N \frac{(a+N)k(\mathbf{V}_\ell, \mathbf{V}_t)}{(a+1)N^2}$$

$$- \frac{2}{Nm} \sum_{\ell=1}^N \sum_{t=1}^m k(\mathbf{V}_\ell, \mathbf{Y}_t) + \frac{1}{m^2} \sum_{\ell,t=1}^m k(\mathbf{Y}_\ell, \mathbf{Y}_t). \quad (8)$$

*Now, it is sufficient to compute the following conditional expectation,*

$$E(\text{MMD}^2_{\text{BNP}}(F_{1,N}^{pri}, F_{2,m})) = E_{H, F_2}(E_{F_1^{pri}}(\text{MMD}^2_{\text{BNP}}(F_{1,N}^{pri}, F_{2,m}) | \mathbf{V}_{1:N})). \quad (9)$$

*Since sets $\{V_i\}_{i=1}^N$ and $\{Y_i\}_{i=1}^m$ include i.i.d. random variables, separately, replacing (8) in expectation (9) implies:*

$$(9) = \frac{a(N-1)}{(a+1)N} E_H[k(\mathbf{V}_1, \mathbf{V}_2)] + \frac{a+N}{(a+1)N} E_H[k(\mathbf{V}_1, \mathbf{V}_1)] - 2 E_{H, F_2}[k(\mathbf{V}_1, \mathbf{Y}_1)]$$

$$+ \frac{m-1}{m} E_{F_2}[k(\mathbf{Y}_1, \mathbf{Y}_2)] + \frac{1}{m} E_{F_2}[k(\mathbf{Y}_1, \mathbf{Y}_1)]. \quad (10)$$

*The proof of (ii) is concluded by letting $a \to \infty$, $N \to \infty$, and $m \to \infty$ in the above equation. Lastly, since $\frac{1}{m} < 1$, $\frac{m-1}{m} < 1$, $\frac{a(N-1)}{(a+1)N} < 1$, and $\frac{a+N}{(a+1)N} < 2$, then, for any $N, m \in \mathbb{N}$ and $a \in \mathbb{R}^+$,*

$$(10) < E_H[k(\mathbf{V}_1, \mathbf{V}_2)] - 2 E_{H, F_2}[k(\mathbf{V}_1, \mathbf{Y}_1)] + E_{F_2}[k(\mathbf{Y}_1, \mathbf{Y}_2)] + 3K,$$

*which concludes the proof of (iii).*

**Lemma 2** *Let $\mathcal{W}$ be the parameter space for $G_{\boldsymbol{\omega}}$ and $\boldsymbol{\omega}^* \in \mathcal{W}$ be the value that optimizes the objective function (2) and $\boldsymbol{\omega}'$ be the true value that minimizes $\text{MMD}(F, F_{G_{\boldsymbol{\omega}}})$. Assume that $F \sim DP(a, H)$ and let $k(\cdot, \cdot)$ be any continuous kernel function with feature space corresponding to a universal RKHS defined on a compact metric space $\mathfrak{X}$ such that $|k(\mathbf{z}, \mathbf{z}')| < K$, for any $\mathbf{z}, \mathbf{z}' \in \mathbb{R}^d$. For a given sample $\mathbf{X}_1, \ldots, \mathbf{X}_n$ from distribution $F$:*
*i. Generalization error:*

$$E(\text{MMD}(F, F_{G_{\boldsymbol{\omega}^*}})) \le \text{MMD}(F, F_{G_{\boldsymbol{\omega}'}}) + \frac{2K}{\sqrt{n}} + \frac{4aK}{a+n} + 2\sqrt{\frac{(a+n+N)K}{(a+n+1)N}}.$$

*ii. Robustness: Suppose there exist outliers in the sample data, which arise from a noise distribution $Q$. Consider the Hüber's contamination model [38, 39], given by $F = (1 - \epsilon)F_0 + \epsilon Q$, where $\epsilon \in (0, \frac{1}{2})$ is the contamination rate, and the latent variables $Z_1, \ldots, Z_n \overset{i.i.d.}{\sim} \text{Bernoulli}(\epsilon)$ are such that $\mathbf{X}_i \overset{i.i.d.}{\sim} F_0$ (cleaned data) if $Z_i = 0$; otherwise, $\mathbf{X}_i \overset{i.i.d.}{\sim} Q$. Then,*

$$E(\text{MMD}(F_0, F_{G_{\boldsymbol{\omega}^*}})) \le \min_{\boldsymbol{\omega} \in \mathcal{W}} \text{MMD}(F_0, F_{G_{\boldsymbol{\omega}}}) + 4\epsilon + \frac{2K}{\sqrt{n}} + \frac{4aK}{a+n} + 2\sqrt{\frac{(a+n+N)K}{(a+n+1)N}}.$$

**Proof of Lemma 2**

The proof of Lemma 2(i) relies on the proof given in [1, Theorem 9] which is expanded for infinite stick-breaking representation, while we consider the finite DP approximation given in (7). By employing a similar technique as in the previously mentioned theorem, we have

$$
\begin{aligned}
E\left(\mathrm{MMD}(F, F_{G_{\boldsymbol{\omega}^*}})\right) = E_F\left(E_{F^{pos}}\mathrm{MMD}(F, F_{G_{\boldsymbol{\omega}^*}})|\mathbf{X}_{1:n}\right) \\
\leq \min_{\boldsymbol{\omega}\in\mathcal{W}}\mathrm{MMD}(F, F_{G_{\boldsymbol{\omega}}}) + 2E_F\left(\mathrm{MMD}(F_n, F)\right) + 2E_{F^{pos}}\left(\mathrm{MMD}(F_N^{pos}, H^*)\right) \\
+ 2E_F\left(E_H(\mathrm{MMD}(F_n, H^*)|\mathbf{X}_{1:n})\right).
\end{aligned}
$$

Building on the results of [1, Lemma 7], we can establish that

$$
E_{F^{pos}}\left(\mathrm{MMD}^2(F_N^{pos}, H^*)\right) \leq \sum_{\ell=1}^{N} E_{F^{pos}}[J_{\ell,N}^{*2}]E_{H^*}[k(\mathbf{V}_\ell^*, \mathbf{V}_\ell^*)] \leq \frac{(a+n+N)K}{(a+n+1)N},
$$

where the right-hand side of the above inequality follows from the fact that $k(\cdot,\cdot) \leq K$ and $E_{F^{pos}}[J_{\ell,N}^{*2}] = \frac{a+n+N}{(a+n+1)N^2}$. Now, the Jensen's inequality implies

$$
E_{F^{pos}}\left(\mathrm{MMD}(F_N^{pos}, H^*)\right) \leq \sqrt{\frac{(a+n+N)K}{(a+n+1)N}}.
$$

On the other hand, [39, Lemma 7.1] and [1, Lemma 8], respectively, imply that

$$
E_F\left(\mathrm{MMD}(F_n, F)\right) \leq \frac{K}{\sqrt{n}}, E_F\left(E_H(\mathrm{MMD}(F_n, H^*)|\mathbf{X}_{1:n})\right) \leq \frac{2aK}{a+n},
$$

which concludes the proof of (i). To establish (ii), we adopt the approach used in the proof of [1, Corollary 5]. Initially, we employ [39, Lemma 3.3] to bound $\mathrm{MMD}(F_0, F_{G_{\boldsymbol{\omega}^*}})$ by $2\epsilon + \mathrm{MMD}(F, F_{G_{\boldsymbol{\omega}^*}})$, resulting in:

$$
E\left(\mathrm{MMD}(F_0, F_{G_{\boldsymbol{\omega}^*}})\right) \leq 2\epsilon + E\left(\mathrm{MMD}(F, F_{G_{\boldsymbol{\omega}^*}})\right).
$$

Applying the result in (i) to the right-hand side of the above inequality implies:

$$
E\left(\mathrm{MMD}(F_0, F_{G_{\boldsymbol{\omega}^*}})\right) \leq 2\epsilon + \min_{\boldsymbol{\omega}\in\mathcal{W}}\mathrm{MMD}(F, F_{G_{\boldsymbol{\omega}}}) + \frac{2K}{\sqrt{n}} + \frac{4aK}{a+n} + 2\sqrt{\frac{(a+n+N)K}{(a+n+1)N}}.
$$

Finally, we employ [39, Lemma 3.3] once again, but this time to bound $\mathrm{MMD}(F, F_{G_{\boldsymbol{\omega}}})$ by $2\epsilon + \mathrm{MMD}(F_0, F_{G_{\boldsymbol{\omega}}})$ for any $\boldsymbol{\omega} \in \mathcal{W}$, thereby completing the proof of (ii).

**Proof of Lemma 1**

Let $\mathcal{L}_{\mathrm{BNP}}(\boldsymbol{\omega}) = \mathrm{MMD}(F_N^{pos}, F_{G_{\boldsymbol{\omega}}})$, $\mathcal{L}_{n,m}(\boldsymbol{\omega}) = \mathrm{MMD}(F_n, F_{G_{\boldsymbol{\omega}},m})$, and $\mathcal{L}(\boldsymbol{\omega}) = \mathrm{MMD}(F, F_{G_{\boldsymbol{\omega}}})$. Then, for $\boldsymbol{\omega}^* \in \mathcal{W}$, [26, Theorem 7] implies

$$
\Pr\left(|\mathcal{L}_{n,m}(\boldsymbol{\omega}^*) - \mathcal{L}(\boldsymbol{\omega}^*)| > h(n, m, K, \epsilon)\right) < 2\exp\frac{-\epsilon^2 nm}{2K(n+m)}. \tag{11}
$$

Hence, with a probability at least $1 - 2\exp\frac{-\epsilon^2 nm}{2K(n+m)}$,

$$
|\mathcal{L}_{n,m}(\boldsymbol{\omega}^*) - \mathcal{L}(\boldsymbol{\omega}^*)| \leq h(n, m, K, \epsilon). \tag{12}
$$

On the other hand, the triangle inequality implies

$$
|\mathcal{L}_{\mathrm{BNP}}(\boldsymbol{\omega}^*) - \mathcal{L}(\boldsymbol{\omega}')| \leq |\mathcal{L}_{n,m}(\boldsymbol{\omega}^*) - \mathcal{L}(\boldsymbol{\omega}^*)| + |\mathcal{L}_{\mathrm{BNP}}(\boldsymbol{\omega}^*) - \mathcal{L}_{n,m}(\boldsymbol{\omega}^*)| + |\mathcal{L}(\boldsymbol{\omega}^*) - \mathcal{L}(\boldsymbol{\omega}')|. \tag{13}
$$

Finally, the proof of (i) is concluded by considering inequality (12) in (13). To prove (ii), Markov's inequality implies

$$
\Pr\left(\mathrm{MMD}(F, F_{G_{\boldsymbol{\omega}^*}}) \geq \epsilon\right) \leq \frac{E\left(\mathrm{MMD}(F, F_{G_{\boldsymbol{\omega}^*}})\right)}{\epsilon}.
$$

The result follows by substituting the bounds from Lemma 4(i) into the right-hand side of the above inequality.

# F  Additional results and Date description

## F.1  Date description

### F.1.1  MNIST

The MNIST dataset includes 60,000 handwritten digits of 10 numbers from 0 to 9 each having 784 ($28 \times 28$) dimensions. This dataset is split into 50000 training and 10000 testing images and is a good example to demonstrate the performance of the method in dealing with the mode collapse problem. We use the training set to train the network.

### F.1.2  BMB [40]:

The bone marrow biopsy (BMB) dataset is a collection of histopathology of BMB images corresponding to 16 patients with some types of blood cancer and anemia: 10 patients for training, 3 for testing, and 3 for validation. This dataset contains 10,800 images in the size of $28 \times 28$ pixels, 6,800 of which are considered for the training set. The rest of the images have been divided into two sets of equal size for testing and validation. The whole dataset can be found at `https://github.com/jmtomczak/vae_householder_flow/tree/master/datasets/histopathologyGray`. The results based on 6800 training images are presented.

### F.1.3  LFD [41]:

The labeled faces in the wild dataset (LFD) include 13,000 facial image samples with 1,024 ($32 \times 32$) dimensions. The dataset is available at `https://conradsanderson.id.au/lfwcrop/`.

## F.2

The sensitivity to the bandwidth parameter of the Gaussian kernel is given by Figure 2.

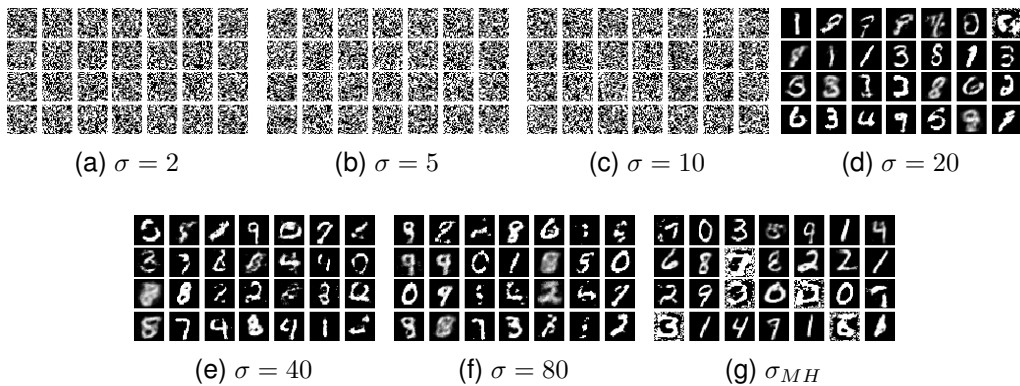

(a) $\sigma = 2$    (b) $\sigma = 5$    (c) $\sigma = 10$    (d) $\sigma = 20$

(e) $\sigma = 40$    (f) $\sigma = 80$    (g) $\sigma_{MH}$

Figure 2: Generated samples from BNPMMD GAN for the MNIST dataset using a single Gaussian kernel with various values of bandwidth parameter $\sigma$ in 40,000 iterations.

## F.3

The learning rate of BNPMMD GAN versus FNP Counterpart is given by Figure 3.

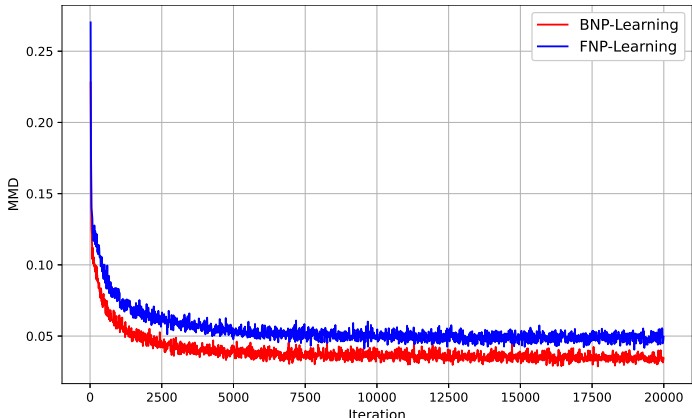

Figure 3: Learning rate: Values of the cost function in the proposed GAN and its frequentist counterpart [22] over $20,000$ iterations.

