# OpenReview forum: "Bayesian Nonparametric Learning using the Maximum Mean Discrepancy Measure for Synthetic Data Generation"
_NeurIPS.cc/2024/Workshop/BDU — NeurIPS BDU Workshop 2024 Poster_

### Official Review · Reviewer_bTmm · 2024-09-19
**Comments of paper #62**

**Rating:** 6
**Confidence:** 3

**Review:**

This paper proposes a Bayesian estimator for maximum mean discrepancy (MMD) and embeds it in a generative adversarial network (GAN), enabling a novel approach to Bayesian nonparametric (BNP) learning. BNP-GAN improves sample diversity and inferential accuracy, outperforming traditional methods.

Pros:
1. A novel scheme for MMD. This scheme embeds a GAN to learn the Bayesian nonparametric networks, which is novel for me.
2. Good writing. This paper is well-written and easy to understand.

Cons:
1. Lacking of sufficient citations for better introduction of the background. For instance, it is suggested to add a related citation within the first statement in introduction “Data augmentation is the technique of generating synthetic data, often to train machine learning models when the data are scarce or the model is non-robust to perturbations in the data.”
2. Unclear statement of dataset selection. In the experimental section, the authors choose several tiny datasets like MNIST and LFW. While I understand that these tiny samples can better fit the Bayesian theoretical proofs, I would like to see more analysis about the relationship between the experimental results from larger data samples like ImageNet and Bayesian lemmas, or theorems. It may be more convincing when proposing the scheme.

---

### Official Review · Reviewer_ZnjJ · 2024-09-26
**Integration of a Bayesian nonparametric (BNP) estimator for the maximum mean discrepancy (MMD) into generative adversarial networks (GANs), with reasonable theoretical support but limited empirical evidence.**

**Rating:** 6
**Confidence:** 4

**Review:**

The paper proposes integrating a Bayesian nonparametric (BNP) estimator for the Maximum Mean Discrepancy (MMD) into the training of generative adversarial networks (GANs) by using it as a discriminator, which is the novelty of this work. The motivation behind this is to address typical challenges with GANs, such as reducing mode collapse and simplifying optimization. The paper provides original theoretical guarantees on the consistency, generalization, and robustness of their estimator, though these may not fully apply in practice due to the complexities of GAN training.

For evaluation, the paper provides preliminary empirical results on datasets such as MNIST. However, the empirical results are limited and do not sufficiently support the advantages of the proposed method compared to frequentist MMD GANs or even regular GANs (see cons below). This raises questions about the significance of the paper, as the empirical results may not support the motivation behind BNP MMD GANs.

## Pros:

- The paper proposes a novel idea of incorporating Bayesian uncertainty into Maximum Mean Discrepancy for GANs. The idea itself, independent of any specific application or use case, is an interesting problem to tackle in Bayesian statistics. The theoretical arguments are sound (though with some concerns raised in the cons) and support the theoretical claims.
- The paper effectively identifies the issue that different data distributions have different uncertainties which cannot be handled by frequentist MMD GAN formulations. It is important to take that into account through a Bayesian formulation.
- The paper is clear and effectively conveys the main idea.
- **L112:** It is commendable that the paper acknowledges that the provided empirical results are preliminary (line 112).

## Cons:

- The empirical evaluation is limited and does not support the claims. In particular:
  - In Table 1, the paper should include the performance of state-of-the-art neural network-based discriminator GANs. It appears that the authors assume MMD GANs outperform such GANs in metrics like FID and KID. However, [1] shows that neural network-based discriminator GANs perform quite well (see Table 3 in Appendix F of [1]). If MMD GANs cannot outperform these models, the only advantage to discuss would be the optimization dynamics of MMD GANs being easier than regular GANs, and the claims on reducing mode collapse may not hold.

  - Regarding the comparison to frequentist MMD methods in Figure 1 and Table 1, the details about the frequentist MMD (FNP) are unclear. Are these results reproduced from prior works? What are the details? It seems that competing methods might perform better than what is shown in the paper based on the literature. For instance, in [2], their MMD GAN generates MNIST samples (see Figure 5.1(c) in [2]) that are visually better compared to the BNP samples in this paper (Figure 1, row 3). This is crucial because if this is the case, then the motivation for improving MMD GANs by incorporating a Bayesian posterior is weakened, and the claims that BNP MMD GANs reduce mode collapse are challenged.

  - FID and KID are random measurements. The paper lacks confidence intervals for their measurements.

  - It would be beneficial to showcase the method on the CIFAR dataset to demonstrate its effectiveness on more complex data.

- While a fine-grained study of the optimization dynamics may be out of scope, the discussion on "better" convergence in **L122-126** is not convincing. First, the experiment is on a mixture of Gaussian distributions. In such a scenario, does a regular GAN have problems with convergence? (or is it difficult to train a regular GAN on such data?) Second, how generalizable is this argument? Would we see the same behavior on MNIST or more challenging datasets? Third, in Figure 3, it appears that both FNP and BNP converge to their local minima at almost the same speed, so it's unclear why the paper claims otherwise. Lastly, since the cost functions of FNP and BNP are different (though both are MMDs), the authors should clarify why comparing different cost functions is informative here.

- In the Lemma 1.ii, the argument is made for $ w^* $. However, in practice, for GANs there is no guarantee of finding the optimal weights due to the non-convexity of the cost function. Therefore, the lemma may not have practical application in the GAN context.

- The theoretical proofs in the appendix feel disjointed. The propositions and theorems are presented without sufficient references or explanations. The whole section is almost an orphan in the main paper, making it difficult to read and connect to the lemmas in the main body. The notation is poorly explained or missing. For instance, what does \( w' \) refer to in **L92**? Moreover, there are issues like referring to Algorithm 2 in **line 110**, while there is no Algorithm 2 in the paper, or referring to Lemma 4 in **line 394**, while there is no Lemma 4.

## References

[1] M. Bińkowski, D. J. Sutherland, M. Arbel, and A. Gretton, “Demystifying MMD GANs” in International Conference on Learning Representations, 2018.
[2] C.-L. Li, W.-C. Chang, Y. Cheng, Y. Yang, and B. Póczos, “MMD-GAN: Towards deeper understanding of moment matching network” Advances in Neural Information Processing Systems, vol. 30, 2017.

---

### Decision · Program_Chairs · 2024-10-09

Accept (Poster)